# Research on Mobile Robot Navigation Method Based on Semantic Information

**DOI:** 10.3390/s24134341

**Published:** 2024-07-04

**Authors:** Ruo-Huai Sun, Xue Zhao, Cheng-Dong Wu, Lei Zhang, Bin Zhao

**Affiliations:** 1College of Information Science and Engineering, Northeastern University, Shenyang 110819, China; sunruohuai@stumail.neu.edu.cn (R.-H.S.);; 2SIASUN Robot & Automation Co., Ltd., Shenyang 110168, China; 3Faculty of Robot Science and Engineering, Northeastern University, Shenyang 110169, China; 4Daniel L. Goodwin College of Business, Benedict University, Chicago, IL 60601, USA; 20203734@stu.hebmu.edu.cn

**Keywords:** SLAM, semantic laser, point cloud, occupation probability

## Abstract

This paper proposes a solution to the problem of mobile robot navigation and trajectory interpolation in dynamic environments with large scenes. The solution combines a semantic laser SLAM system that utilizes deep learning and a trajectory interpolation algorithm. The paper first introduces some open-source laser SLAM algorithms and then elaborates in detail on the general framework of the SLAM system used in this paper. Second, the concept of voxels is introduced into the occupation probability map to enhance the ability of local voxel maps to represent dynamic objects. Then, in this paper, we propose a PointNet++ point cloud semantic segmentation network combined with deep learning algorithms to extract deep features of dynamic point clouds in large scenes and output semantic information of points on static objects. A descriptor of the global environment is generated based on its semantic information. Closed-loop completion of global map optimization is performed to reduce cumulative error. Finally, T-trajectory interpolation is utilized to ensure the motion performance of the robot and improve the smooth stability of the robot trajectory. The experimental results indicate that the combination of the semantic laser SLAM system with deep learning and the trajectory interpolation algorithm proposed in this paper yields better graph-building and loop-closure effects in large scenes at SIASUN large scene campus. The use of T-trajectory interpolation ensures vibration-free and stable transitions between target points.

## 1. Introduction

The significance of simultaneous localization and mapping (SLAM) technology and trajectory interpolation for mobile robots and autonomous driving has been increasing due to the continuous development of artificial intelligence technology [1,2,3,4,5,6]. SLAM algorithms and trajectory interpolation have been successfully applied in various fields, including campus inspection, logistics and distribution, and unmanned driving. When dealing with large-scale outdoor environments such as factories, laser point clouds are less affected by weather and light and can perceive a 360-degree range. However, it is essential to note that their operation speed is faster. However, laser point clouds typically only contain the geometric information of the environment. In dynamic environments, they may generate residual shadows on the map, which can decrease the accuracy of laser mapping [7,8]. Loopback detection in laser SLAM relies on traditional features such as position and intensity. However, this method could be better as it needs to consider the semantic information of the surrounding environment, which is crucial for human beings to recognize whether a place has been reached [9,10].

To address the issues, this paper explores a semantic laser SLAM system that incorporates deep learning and a trajectory interpolation algorithm. Compared to existing methods, the SLAM system presented in this paper incorporates the concept of voxels into the occupation probability map, thereby improving the ability of local voxel maps to represent dynamic objects. This paper combines the deep learning semantic laser SLAM system with trajectory interpolation algorithm research to solve the system problem in which the sensors cannot directly obtain the semantic information of the point cloud and recognize the points on the dynamic objects. The related algorithms of the deep learning point cloud semantic segmentation are also incorporated. The article utilizes a PointNet++ network for point cloud semantic segmentation. This network can recognize points on dynamic objects and extract deep features of a scene’s dynamic point cloud to output semantic information of points on static objects. A global environment descriptor containing semantic information is generated to identify loops and add loop constraints to the factor graph for optimization. By adding semantic information, dynamic points in the map can be filtered out to improve the map-building quality. Map building and trajectory interpolation experiments were conducted on the SIASUN Campus. The experimental results were compared with satellite maps to demonstrate the algorithm’s ability to build maps and localize in large scenes. The robot’s planning curves show that T-trajectory optimization effectively ensures vibration-free and stable transitions between target points. The experiments demonstrate the feasibility of the proposed algorithm. The time-consuming analysis of the SLAM system shows that it can perform real-time computation, meeting the real-time localization requirements of mobile robots.

The main contributions of this paper are as follows:(a)This study proposes a mobile robot navigation method based on semantic information, combining a semantic laser SLAM system based on deep learning and a trajectory interpolation algorithm to solve the navigation challenges in dynamic environments and large scenes.(b)This study introduces the concept of voxels into occupancy probability maps to better represent dynamic objects. The PointNet++ network is used for point cloud semantic segmentation to identify dynamic object points and extract semantic information.(c)This study uses semantic information to generate a global environment descriptor for loop detection and map optimization, improving the quality of map construction.(d)This study proposes a T-trajectory interpolation algorithm to ensure the smooth transition of robot motion and avoid vibration.(e)This study was experimentally verified in the SIASUN campus environment. The SLAM system can achieve real-time calculation and meet the positioning needs of mobile robots.

## 2. Mobile Robotic Systems Overview

### 2.1. Mobile Robot Navigation Technology Program

As depicted in Figure 1, several open-source laser SLAM algorithm frameworks are available [11,12].

1.Featured-Based Registration:

LOAM (Lidar Odometry and Mapping in real-time):

Front-end odometry: Scan to scan—feature point-based alignment L-M nonlinear optimization.

Back-end optimization: Scan to map—map optimization.

LEGO-LOAM (Lightweight and Ground-Optimized Lidar and Mapping):

Front-end odometry: Point cloud segmentation, scan to scan—two-stage L-M nonlinear optimization for feature point-based alignment.

Back-end optimization: Scan to map–map optimization, graph optimization, and loopback detection.

LIO-SAM:

Front-end odometer: Front-end odometer fused with IMU.

Back-end optimization: Graph optimization with added GPS factor and loopback detection.

2.Direct Registration:

Featured-Based Registration:

This approach to image registration relies on the minimization of some distance or dissimilarity metric between the target image and the input/moving image. Common distance measures employed include the Sum of Squared Differences (SSD), Sum of Absolute Differences (SAD), and Normalized Cross-Correlation (NCC). These distance metrics quantify the similarity between the two images by evaluating the pixel-wise intensity differences. By optimizing these distance measures, the optimal geometric transformation (e.g., translation, rotation, scaling) between the two images can be estimated. Distance-based registration methods are computationally efficient but tend to be sensitive to noise, intensity inhomogeneities, and partial occlusions in the images.

Local Distribution-Based Registration:

This class of image-registration methods based on probability distributions focuses on aligning the underlying probability distributions of the two images, rather than simply comparing pixel-level intensity differences. Common distribution similarity measures employed include Mutual Information (MI) and Normalized Mutual Information (NMI). These measures quantify the statistical correlation between the intensity values in the two images, capturing the similarity of their joint probability distribution and marginal probability distributions.

Compared to distance-based registration methods, distribution-based approaches are generally more robust to imaging artifacts such as noise, intensity inhomogeneities, and partial occlusions. This is because they focus on the underlying statistical properties of the image features, rather than just pixel-wise intensity differences.

However, distribution-based registration methods are typically more computationally complex, requiring sophisticated probability density estimation and optimization procedures. In some applications, their convergence may be slower. Therefore, in practice, hybrid strategies combining both distance-based and distribution-based methods are often employed to achieve optimal registration performance.

### 2.2. General Framework of the SLAM System

SLAM algorithms are essential for enhancing the autonomy and intelligence of mobile robots. Recent research has identified two types of laser SLAM systems: feature-based alignment and direct alignment based on point cloud alignment methods. This paper proposes a framework for the SLAM system, as shown in Figure 2, which receives inputs from 3D LIDAR and outputs 6 DOF attitude estimates. The system is divided into three modules: front-end odometry, back-end nonlinear optimization, and loopback detection.

The front-end odometers infer rough radar motion from adjacent frames of radar data and provide initial values for the back end. Front-end alignment methods include ICP matching, NDT matching, PL-P matching, and CSM matching. ICP matching utilizes point cloud data to construct local geometric features, NDT matching constructs multidimensional variables based on normal distributions, PL-P matching approximates the actual surfaces using a segmented linear method, and CSM matching obtains the initial values through correlation scanning and least squares problems. The nonlinear optimization methods include gradient descent, the Gaussian Newton method, and the L-M method, where the L-M method introduces the trust region based on the Gaussian Newton method.

Loopback detection, also known as closed-loop detection, enables robots to recognize previously visited locations and achieve closed-loop capabilities for mapping. There are various methods for loopback detection, including feature matching, odometer position-based, and deep learning methods. Additionally, radar data can be matched using Scan-to-Scan, Scan-to-Map, and Map-to-Map methods.

Back-end optimization aims to improve the accuracy of estimating the robot’s previous bit positions and waypoints in the presence of noise by reducing estimation errors in motion states and waypoints. The process of state prediction and measurement updating involves modeling the robot’s motion and applying Kalman filtering. Additionally, graph-based optimization methods are used to represent the robot’s poses as variables to be optimized and construct a graph of the error terms through the relationships between the poses. These methods effectively improve the accuracy of robot localization and map construction in complex environments.

## 3. Map Organization and Update Strategy

### 3.1. Laser Odometry—Nonlinear Optimization Algorithm

Navigation systems use odometry data to estimate changes in robot position over time. In SLAM problems, both the front-end position optimization problem and the back-end graph optimization problem are modeled as nonlinear least squares problems. Therefore, nonlinear optimization algorithms are crucial for SLAM systems. A general nonlinear least squares problem can be defined with a minimization objective function.
(1)$$\underset{x}{min}\;F(x)=\underset{x}{min}\frac{1}{2}∥f(x){∥}^{2}$$
where $x\in R^{n},\;f(x)$ is a nonlinear function.

### 3.2. Voxel-Based Local Map Construction and Updating

Compared to the 2D case, building and utilizing a 3D raster map for point cloud alignment requires significantly more computational effort, leading to an increased burden on the system and reduced real-time performance. We utilize NDT maps to address the issue of aligning 3D point clouds. These maps employ the distribution of points within a voxel to represent the entire voxel. The NDT algorithm divides the voxel into relatively large sections, treating the points within each voxel as sampling points of a single Gaussian distribution. The mean and covariance of the distribution of points within the voxel are then fitted. Additionally, rasters are utilized in the NDT algorithm to divide the map.

The paper introduces a voxel-based map representation for alignment, where local maps are voxelized, and occupancy probability is incorporated to enhance the representation of dynamic objects. The environment is modeled using 1 m^3^ square voxels to match the outdoor scene. Figure 3 illustrates the voxel mapping process. When aligning with the local map for the current frame, it is typically necessary to find the nearest neighbor of the point or voxel raster to establish the alignment relationship. Utilizing the hash algorithm for local voxels can expedite the voxel-finding process and simplify voxel addition and deletion operations, resulting in lower algorithmic complexity.

In the figure, m and c represent the mean and covariance, respectively, n represents the number of measurement points, and pk represents one of the measurement points.

When a new sampling point enters the voxel, the distribution of the voxel is corrected using an iterative update strategy. The traditional method of iterative updating each time is computationally expensive compared to this scheme, which significantly reduces computation. This correction scheme for map voxel information is a primary research focus for maps. The original mean and covariance will not reflect the current new distribution within the voxel when a new point enters the voxel. Incremental corrections can be made to the existing mean and covariance through an iterative approach rather than re-calculating them for all points. The data structure of the voxels in this system includes the mean, covariance, and occupancy probability as core parameters.

## 4. Combining Deep Learning for a Semantic Laser SLAM System

This paper enhances the network structure of previous research. First, a multi-layer feature extraction module is used to achieve deep learning-based segmentation of dynamic object point clouds. Additionally, the output layer of the network is modified and retrained to output the semantic categories of static object points. Second, the neural network’s semantic results create a global environment descriptor based on semantic information. Geometric and semantic similarity matching is used to identify loopback candidates, and then map-to-map matching is employed to customize the loopbacks precisely. This prevents the addition of erroneous constraints to the factor graph. Finally, in the outdoor environment with dynamic objects, we construct a pure 3D laser semantic SLAM algorithm to filter dynamic points based on semantic information and create a static semantic map of the point cloud. We generate a global environment descriptor containing semantic information and detect loopbacks using the loopback detection method, which is then optimized using the factor graph.

### 4.1. Segmentation Feature Extraction Based on Ground Constraints

Point cloud semantic segmentation is crucial in 3D applications, as it provides high-precision localization information for SLAM systems to construct accurate maps. Additionally, it offers reference targets for buildings and man-made features in building information models.

A neural network framework-based scheme for semantic segmentation of 3D point clouds can determine the object categories in the point cloud data and provide a more comprehensive description of the environmental scene. This paper applies the PointNet++ point cloud semantic segmentation network to outdoor large scene point clouds with an uneven density and a large data volume. PointNet++ processes a set of points sampled in the metric space by building a multilayer neural network and extracts the features of the sampled points through multiple simplified PointNet [13]. As illustrated in Figure 4, PointNet++ comprises multiple Set Abstraction (SA) layers. For each SA layer, the input vector is either the original point cloud or the local features extracted from the previous SA layer. The features of each layer are extracted using PointNet and then combined by a combination layer in the next SA layer to extract deeper features.

The Set Abstraction Layer comprises three main components: the Sampling Layer, the Grouping Layer, and the PointNet Layer. The Sampling Layer selects a set of points from the input point set to serve as the center of the local neighborhood. The Grouping Layer constructs the local point set, which defines the local region of the centers. The PointNet Layer uses a mini PointNet to encode the local point set and obtain the feature vectors.

### 4.2. Closed-Loop Detection and Position Optimization Flow

Closed-loop detection is a crucial component of the laser SLAM system. It ensures map consistency and eliminates accumulated errors during point cloud alignment, particularly when building maps for large scenes. The closed-loop detection strategy of the laser SLAM system can be divided into two algorithms: descriptor-based closed-loop detection and positional nearest neighbor-based closed-loop detection. The descriptor-based detection algorithm compresses high-dimensional point cloud data by extracting features from the point cloud. By comparing the low-dimensional descriptor data of two frames of the point cloud, it can be quickly determined whether they may have been sampled from the same scene. The closed-loop nearest-neighbor detection algorithm compares the error values between the descriptors of the laser point cloud of the current frame and the descriptors of the point cloud of the historical laser keyframes. The worst and most minor historical point cloud descriptors are then selected to obtain the most probable closed-loop point of the current position.

The process of closed-loop detection, also known as scene recognition, involves generating a global environment descriptor that contains semantic information, which is then used for scene description and search. Once the closed loop is successfully detected, the bit positions in the global keyframes are optimized through graph optimization. The closed-loop detection thread can perform this step separately to complete the global map optimization and reduce cumulative errors. The system extracts the local sub-map from the new global map and uses it to recreate the voxel map, completing the update operation of the old local map.

## 5. T-Trajectory Interpolation Strategy

The SLAM system enables the robot to accurately localize and build a map in unknown environments. Additionally, the trajectory interpolation feature generates smooth paths, ensuring the smooth motion of the robot. In the manipulation space, paths and poses are planned and interpolated separately. The resulting per-cycle positions are solved by inverse kinematics based on the model to obtain the corresponding joint angles for motion control. T-trajectory interpolation is designed to ensure that the robot exhibits smooth, accurate, and efficient motions when executing T-tracks. T-trajectory interpolation is the process of generating and optimizing T-trajectories in a robot, CNC machine, or other automation system. It involves inserting additional points into the path of the robot to ensure smooth, accurate, and efficient motion.

The objective of T-trajectory interpolation is to enable the robot to display desirable motion characteristics while executing T-trajectories using suitable mathematical algorithms and control strategies.

Acceleration:(2)$$A\left(t\right)=\left\{\begin{array}{cc} A & 0⩽t<t_{1}\\ 0 & t_{1}⩽t<t_{2}\\ -A & t_{2}⩽t<t_{3} \end{array}\right.$$

Speed:(3)$$V(t)=\left\{\begin{array}{cc} A\tau_{1} & 0\leq t<t_{1}\\ AT_{1} & t_{1}\leq t<t_{2}\\ AT_{1}-A\tau_{3} & t_{2}\leq t<t_{3} \end{array}\right.$$

Distance:(4)$$S(t)=\left\{\begin{array}{cc} S_{s}+\frac{1}{2}A\tau_{1}^{2} & 0\leq t<t_{1}\\ S_{01}+AT_{1}\tau_{2} & t_{1}\leq t<t_{2}\\ S_{02}+AT_{1}\tau_{3}-\frac{1}{2}A\tau_{3}^{2} & t_{2}\leq t<t_{3} \end{array}\right.$$

In Equations (2)–(4), A represents the acceleration of T-trajectory interpolation, *t*_1_~*t*_3_ represent the time nodes of the three-segment planning respectively, and $\tau_{1}\sim \tau_{3}$ represent the time elapsed in the three segments. $S_{s}$ represents the distance traveled at the beginning of interpolation, $S_{01}$ represents the cumulative distance traveled in the first segment of interpolation, and $S_{02}$ represents the cumulative distance traveled in the second segment of interpolation.

T-trajectory interpolation is a technique that helps to prevent robot instability when switching paths. Its key features include:Smooth transitions: Ensuring that the transitions of the robot between connecting target points are smooth to avoid erratic motion.Trajectory Optimization: Interpolation algorithms can be used to generate T-trajectories that optimize the trajectory for a given motion condition, ensuring the shortest path, minimum acceleration/deceleration, and minimum mechanical stress.Velocity Planning: The interpolation algorithm must consider the velocity changes in each part of the T-trajectory to maintain system stability by avoiding excessive speed or slowness.

The acceleration A is determined by the drive of the chassis motor and the friction between the tire and the ground. The maximum speed of the T-trajectory interpolation is determined by the vehicle’s movement capability and cannot exceed the vehicle’s maximum speed. The time of each segment of T-trajectory interpolation is determined by the running speed and walking distance.

## 6. Experimental Results and Analysis

A common use case involves a vast industrial complex located in SIASUN, comprising of fixed structures (static features), parked vehicles (semi-static features), pedestrians, and moving vehicles (dynamic features) in a typical dynamic environment. Figure 5 showcases an outdoor experimental vehicle system that verified the proposed navigation architecture in this paper on the SIASUN Smart Park campus. The experiment was based on the outdoor security inspection robot developed by SIASUN. The robot is a four-wheeled ground mobile vehicle with an Ackerman structure and is equipped with sensors such as LIDAR, camera, GPS, and IMU.

The study used a 3D LiDAR as a data source and an RTK-GPS system, which was constructed using GPS combined with a self-built base station, to provide trajectory truth data for the experiment. Table 1 displays the specifications of the equipped LiDAR model, the Sprint 16-line LiDAR RS-LiDAR-16.

### 6.1. Trajectory Interpolation Test

First, the curves of the mobile robot for trajectory interpolation are collected and tested to verify the key features such as smooth transition, trajectory optimization, and velocity planning.

The aim of T-trajectory interpolation is to achieve optimal motion characteristics by utilizing appropriate mathematical algorithms and control strategies when executing T-trajectories. This improves the accuracy and efficiency of the automation system. Figure 6a–c demonstrate that T-trajectory interpolation results in a vibration-free and stable transition between target points, preventing robot instability during path switching. Trajectory optimization shapes the trajectory to meet specific motion conditions, considering factors such as the shortest path, minimum acceleration/deceleration, and minimum mechanical stress. Velocity planning ensures appropriate velocity variations in different parts of the T-shaped trajectory to avoid excessively fast or slow movements, thereby improving system stability.

### 6.2. Semantic Maps and Closed-Loop Detection Experiments

The robot motion control system used in the experiment was independently developed by SIASUN. The weights obtained by training the pointNet++ network with the KITTI dataset were imported into the vehicle-mounted NVIDIA 3090 PC. In the experiment at SIASUN Park, the point cloud generated by the vehicle-mounted 3D laser was input into the model, and the dynamic objects in the map were filtered out according to the classification results to construct the map.

The experiments on graph building were conducted using the ALOAM algorithm, and our algorithm on the KITTI dataset sequence 05. The resulting trajectories were compared to the true values, and the results are presented in Figure 7a,b.

Figure 7a displays the original semantic map without dynamic point filtering. It is evident that the blue section represents the residual shadow left by dynamic objects, which negatively impacts the quality of the map. By contrast, Figure 7b shows the map generated after dynamic object filtering during construction. The blue dynamic points have been filtered out, resulting in a reduction in the number of dynamic points in the map and an improvement in its quality. The conventional SLAM laser algorithm that uses feature point cloud building is unable to process dynamic points, which leads to residual shadows in the generated map.

The point cloud to be processed is projected onto the voxel, the green dynamic voxel in the figure is found according to the occupancy probability, then the dynamic voxel is dilated, and finally the point cloud to be processed is projected onto the dynamic voxel, and all points belonging to the dynamic voxel are removed.

As shown in Figure 8, from the horizontal comparison between our algorithm and the A-LOAM algorithm, it can be seen that the algorithm effectively removes the dynamic points belonging to vehicles and pedestrians.

Figure 9 shows the final global path of the robot, which has the same start and end points, indicating that the algorithm can effectively detect closed loops. Experimental tests on the SIASUN C1 building dataset demonstrate that the system can accurately identify closed-loop constraints and perform graph optimization. The SIASUN C1 Building Dataset is a point cloud dataset captured using a three-dimensional LiDAR sensor by scanning the main building within the SIASUN Industrial Campus. The data were acquired by circling the perimeter of the primary building structure.

The experiments demonstrate that the combination of a semantic laser SLAM system with deep learning and a trajectory interpolation algorithm can effectively utilize semantic information to identify loopbacks and perform global graph optimization, resulting in reduced cumulative errors and smoother robot trajectories.

### 6.3. Large-Scale Mapping Experiment for a Corporate Campus

The SIASUN outdoor mobile robot platform was used to extensively survey the periphery of the SIASUN campus. The effectiveness of whole-map building in a large scene was analyzed in this experiment. The first and last trajectories were connected to complete the experiment. An outdoor security inspection robot was used to patrol the periphery of the SIASUN campus, as depicted in Figure 10a. The route covered a total distance of approximately 1.8 km, forming a long-distance loop. The environment consisted of typical outdoor features such as trees, buildings, and open roads. The robot’s specific route is depicted by the red curve in Figure 10b. The global map, built by the robot, is shown in yellow-green, while the relative position of the trajectory calculated by the algorithm and the built map is shown in red.

Figure 10a,b show that the point cloud map aligns accurately with the satellite map, including the building edges, road edges, and tree shadows. The color change in the point cloud map reflects the height difference and confirms the color block change in the grayscale map. The results indicate that the proposed SLAM system achieves high positioning accuracy while maintaining good mapping efficiency. In Figure 10c, the brown trace depicts the motion trajectory traversed by the robot during the data collection process. Additionally, the distance values at various points along the path are also illustrated. An analysis of the provided results reveals that the trajectory data generated by the algorithm in this work exhibit a high degree of spatial alignment with the reference satellite map data. Notably, the computed trajectory does not exhibit any abrupt or discontinuous changes in the path. This indicates that the employed T-trajectory interpolation approach has successfully maintained the stability and continuity of the motion profile throughout the evaluated operation.

## 7. Conclusions

This paper presents a complete solution for SLAM systems by combining a semantic laser SLAM system with deep learning and a trajectory interpolation algorithm. The work includes the following points:(1)This paper proposes a general framework for a SLAM system based on open-source laser SLAM algorithms.(2)The NDT algorithm addresses the issue of aligning 3D point cloud alignments. It employs the feature point method for feature extraction and scan-to-map alignment of the point cloud to obtain the robot position with high accuracy. This enhances the ability of local voxel maps to represent dynamic objects.(3)The semantic categories of the points are labeled as the point cloud and are dynamically segmented using PointNet++. A global environment descriptor is generated based on the semantic information, and loopbacks are detected using a loopback detection method. The loopbacks are then optimized using a factor graph.(4)The SLAM navigation algorithm employs T-trajectory interpolation for global and local planning to ensure the performance of the robot motion, resulting in a smooth and stable trajectory.

Experiments demonstrate that the semantic laser SLAM system can accurately recognize semantic information of points on both moving and static objects, meeting the basic requirements of the SLAM system in terms of operational speed. Combining the deep learning semantic laser SLAM system with the trajectory interpolation algorithm reduces cumulative errors and provides a solid foundation for generating high-precision maps. The deep learning algorithm was tested on public datasets and compared with other SLAM algorithms. The results demonstrate that this algorithm satisfies the requirements of SLAM algorithms and is practical and feasible in outdoor scenes with dynamic objects.

## Figures and Tables

**Figure 1 sensors-24-04341-f001:**
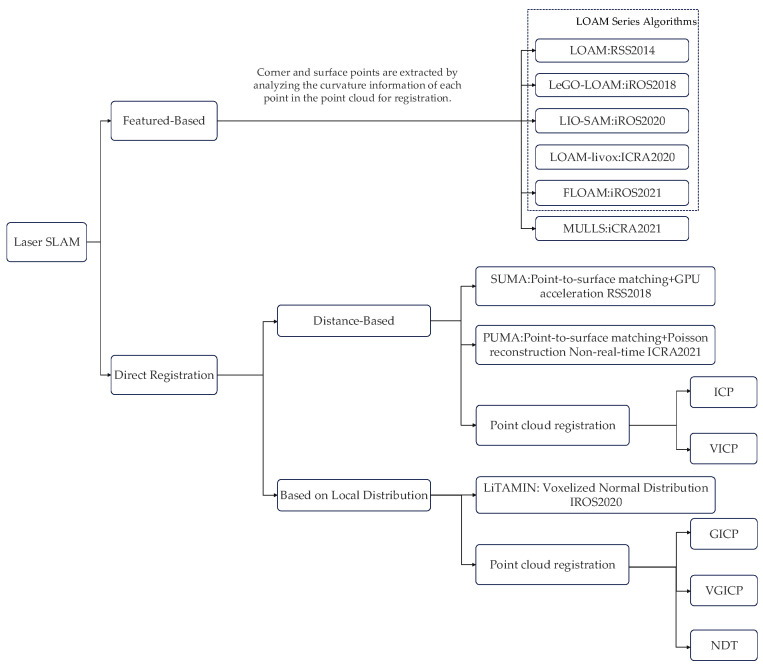
Laser SLAM system.

**Figure 2 sensors-24-04341-f002:**
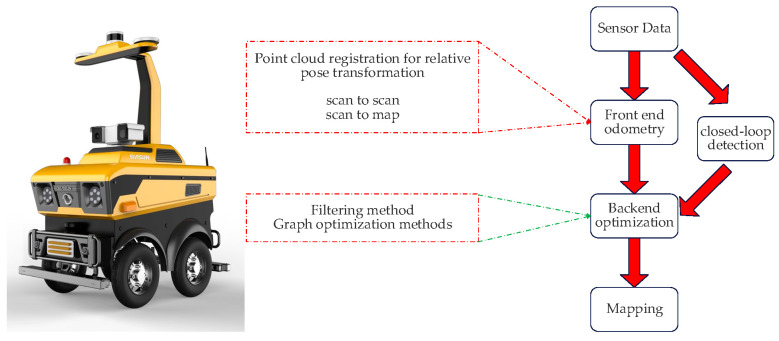
Robotic laser SLAM software platform.

**Figure 3 sensors-24-04341-f003:**
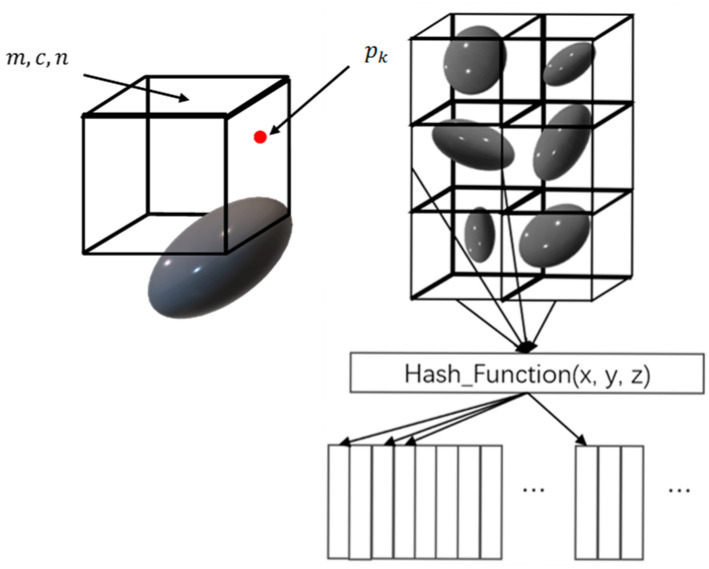
Schematic of voxel mapping.

**Figure 4 sensors-24-04341-f004:**
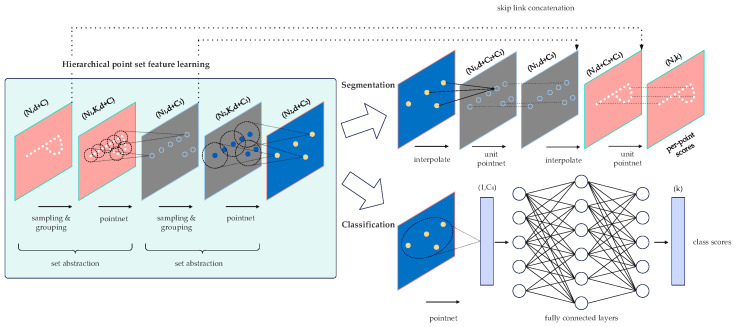
Schematic diagram of PointNet++ network structure.

**Figure 5 sensors-24-04341-f005:**
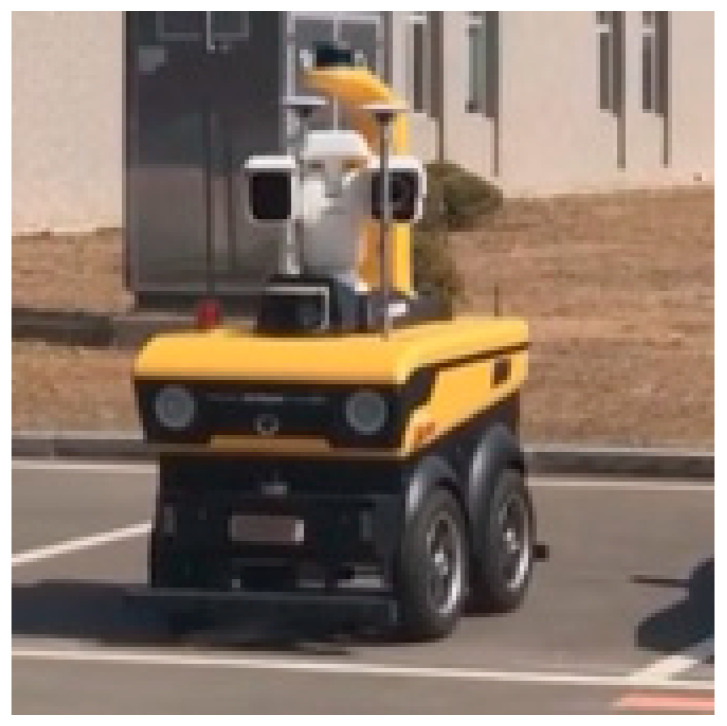
Outdoor experimental vehicle system.

**Figure 6 sensors-24-04341-f006:**
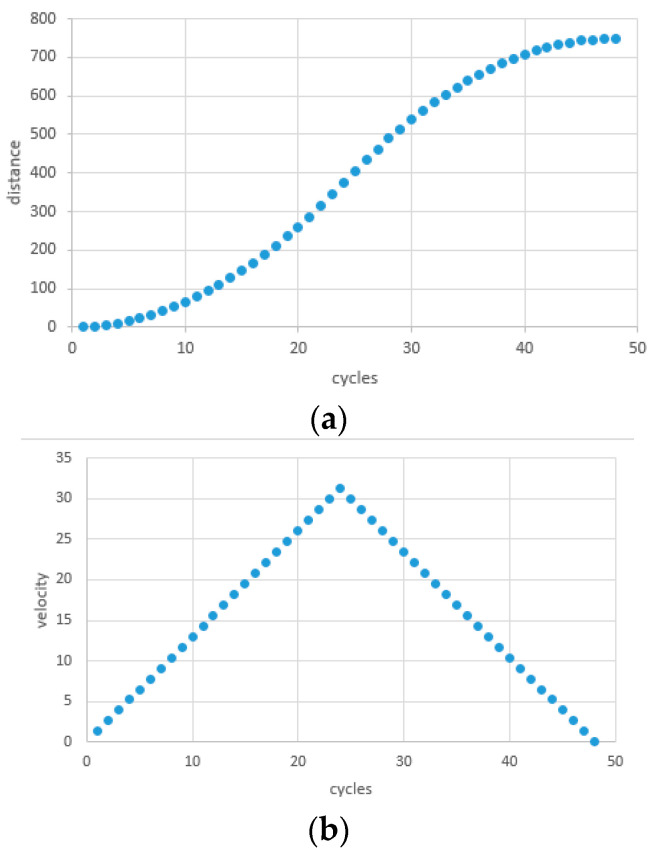
(**a**) Location curve for T planning. (**b**) Velocity curve for T planning. (**c**) Acceleration curve for T planning.

**Figure 7 sensors-24-04341-f007:**
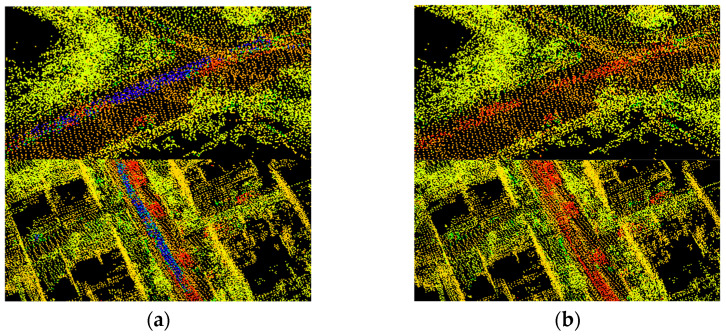
(**a**) Original semantic map without dynamic point filtering. (**b**) Static semantic map after dynamic point filtering.

**Figure 8 sensors-24-04341-f008:**
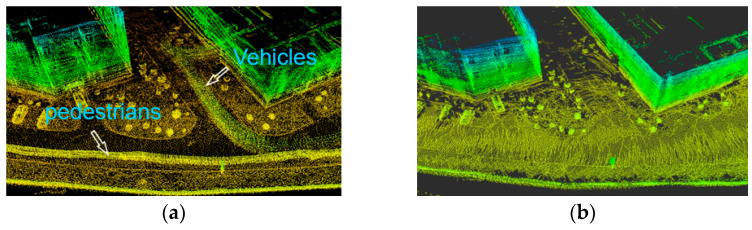
(**a**) Dynamic point filtering effect of A-loam algorithm. (**b**) Dynamic point filtering effect of proposed algorithm.

**Figure 9 sensors-24-04341-f009:**
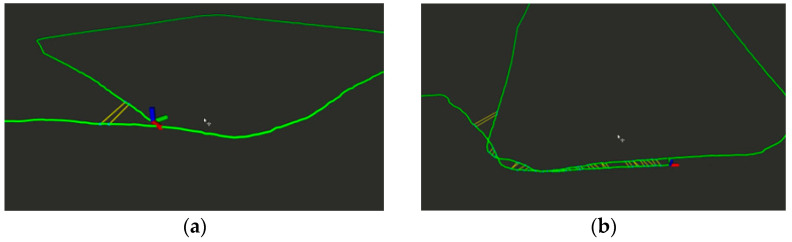
(**a**) Closed-loop trajectory from the horizontal perspective. (**b**) Closed-loop trajectory from the bird’s-eye view perspective.

**Figure 10 sensors-24-04341-f010:**
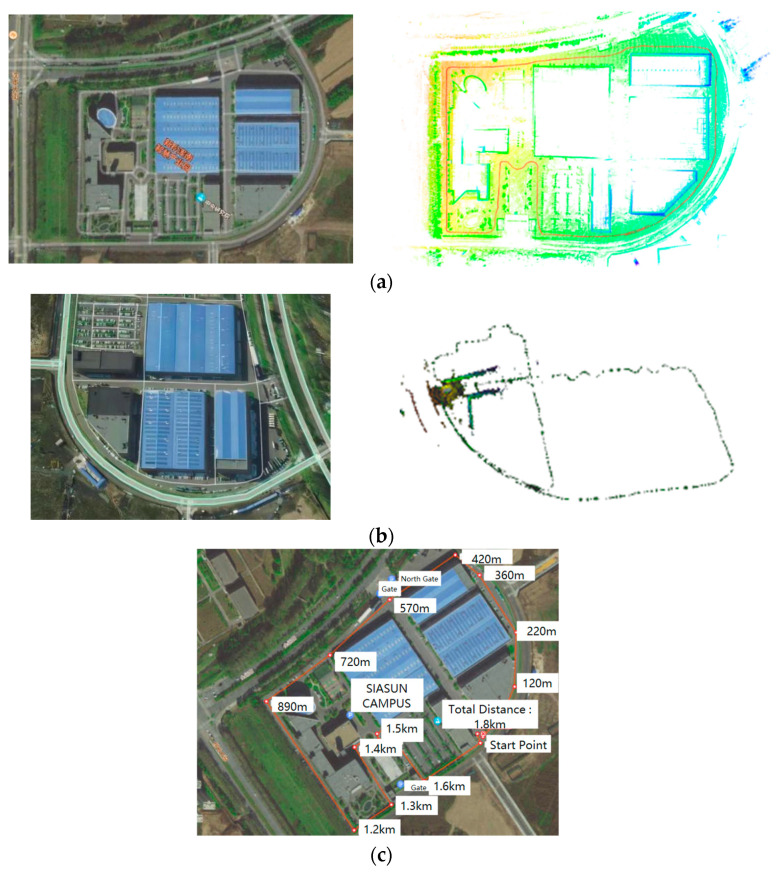
(**a**) Global mapping outcomes and associated trajectory. (**b**) Robot traversal trajectories during the mapping process. (**c**) Robot’s motion trajectories during the data acquisition process.

**Table 1 sensors-24-04341-t001:** RS-LiDAR-16 parameter specifications.

Parameter	Specification
Horizontal field of view	360°
Vertical field of view	30°
Horizontal angular resolution	0.1°/0.2°/0.4°
Frame rate	5 Hz/10 Hz/20 Hz
Ranging capability	150 m
Accuracy (typical)	±2 cm

## Data Availability

Data are contained within the article.

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
