# Peer review of "Research on Mobile Robot Navigation Method Based on Semantic Information"

_sensors, 2024, doi:10.3390/s24134341_

Round 1

Reviewer 1 Report

Comments and Suggestions for Authors

The author proposed a solution for the problem of mobile robot navigation and trajectory interpolation in dynamic environments with large scenes. I believe the content of the author's research is comprehensive and logically rigorous, constituting a complete work on mobile robot navigation, involving SLAM and trajectory planning. However, I think there are still some questions that the author needs to address:

1.     Innovation: Where does the author's innovation lie, considering that the mobile robot navigation framework proposed by the author is based on an open-source SLAM system, and the NDT point cloud registration algorithm used by the author is also a commonly used algorithm? There is related work on the study of dynamic object filtering. Where is the author's innovation?

2.     Format issues: There are some simple format issues in the article that need to be addressed. Firstly, on line 47, the use of "Unfortunately" may be inappropriate. Additionally, in the Introduction, content from line 52 to 57 is repetitive. In the Overview section, the literature citations for the open-source SLAM framework should correspond to specific SLAM frameworks. The format of "1m3" on line 157 should be modified, and the format of equations 2-4 should also be revised.

3.     Detail issues: The author's introduction to the use of T-trajectory interpolation is too brief. Is the A coefficient set as a constant or does it dynamically change? Additionally, how does this algorithm ensure the shortest and optimal trajectory?

4.     Experiment issue: The author's research focus is on the semantic LiDAR SLAM system, but the experimental results do not reflect the semantic information. The author needs to supplement relevant experimental results to substantiate this.

Overall, I believe this article is rich in content and has a clear structure, making it acceptable. However, the aforementioned issues need to be addressed by the author.

Reviewer 2 Report

Comments and Suggestions for Authors

This paper proposes a semantic laser SLAM combined with a trajectory interpolation method for mobile robot navigation based on the SLAM systems. Although the contents of this paper may be useful for the mobile robot navigation, the description is insufficient to understand the proposed method. The authors need to clarify what kind of problem they have addressed by introducing a novel approach of the proposed method. Therefore, additional descriptions and reconstruction of the structure are needed in this paper to improve the quality of the presentation.

Some comments for the authors are the following:
L72 & Figure 4:
You should appropriately cite the article of PointNet++ (Qi et al., 2017).
Additionally, I think that Figure 4 in p.6 is similar to an illustration posted on the site of PointNet++ (https://github.com/charlesq34/pointnet2)(corresponding to Figure 2 of the article of PointNet++). You should correctly cite the related papers as references or replace the figure.

Figure 1 in p.3:
A description of Figure 1 is not sufficient in text. There is only a partial explanation of the LOAM algorithms. In particular, there is no explanation of the direct registration in text of section 2.1. You should add the description of the algorithms shown in the figure. If not, the figure should be revised to a description corresponding to text.

Figure 3 in p.5 & Eqs. (2),(3),(4) in p.7:
Symbols of variables and constants of Figure 3 and Eq. (2)(3)(4) are not described in clear explanation in text. It is necessary to add a description of the symbols for understanding the proposed method.

Sections 4,5,6:
This paper does not appropriately describe how to implement the algorithms to the developed system in text. You should add more detailed descriptions of the architecture of the developed SLAM system and how to build a deep learning model by PointNet++ architecture for segmentation and classification of point clouds.

Section 6 in p.7:
It is recommended to add information on the experimental setup. It is difficult to understand the experimental results because of insufficient robot information. You should describe more detailed hardware configuration other than LiDAR and the software configuration of the robot and show the proposed framework of SLAM including the filtering of the dynamic points by semantic information and the T-trajectory interpolation process.

Figure 7 in p.9:
Figure 7 is insufficient in information to appreciate the results. Currently, it is unclear whether the results are by the proposed method, ALOAM, or Lego-LOAM. There is one thing for sure that the blue section of dynamic points in Fig. 7 can be filtered out. It is recommended to add the necessary information for appreciating the results.

Figure 8 in p.10:
If you use an original dataset (SIASUN C1 building dataset), you should explain more detailed information about the dataset. Moreover, from only a graph of Fig. 8, it is not possible to estimate the effect of the closed loop detection. You should add results without closed loop detection to verify the accuracy of the trajectory path and evaluate more quantitatively the reduction of cumulative errors and smoother robot trajectory.

Section 5.3:
It is difficult to understand the result of section 5.3 because of insufficient in information. Is the operation of the robot automatic or manual? In Figure 9, where are the start and end positions of the robot? etc.

In section 5.3 or the previous section, you should describe more detailed information about the Large-Scale Mapping Experiment to appreciate the results. Only these graphs of Fig. 9 are not sufficient for quantitative evaluation. It is recommended to add a quantitative evaluation of the robot's trajectory compared with data from RTK-GPS. Moreover, if possible, it is recommended to add comparison results without the T-trajectory interpolation method to clearly demonstrate the effect of the T-trajectory interpolation.

Minor comments:
L47 to L61: The sentences are the same as l34 to l47. You should eliminate the duplication and correct this paragraph.

L92 (Lidar et al.): Reference should be correctly cited.

Eqs (2),(3),(4): It is recommended to add the space between equation and condition

section 5.1, 5.2 5.3, and 6. Patents: Section numbers are confusing. It is recommended to fix section numbers in the right order.

Figure 6: It is recommended to add the label of each axis in Figure.

Figure 9: Figure style (background colors) should be matched between figures.

L387 to L409: The journal titles of references are incorrect. They should be corrected.

Reviewer 3 Report

Comments and Suggestions for Authors

The article presents a mobile robot navigation method using SLAM and deep learning, which the authors use to detect dynamic objects. The results are presented in key figures 7 and 8. There are some shortcomings in the article that need to be addressed (see comments).

Comments:

1. Lines 34-47 are repeated in lines 48-61

2. Arrange equations 2, 3, 4 to be readable by spaces between values and conditions.

3. Complete the descriptions of the axes in fig. 6, add the units on the axis in fig. 8.

4. Subsection numbers in section 6 begin with the number 5.

5. There is no comparison of the results with other published works.

6. Considerably expand the list of references.

Round 2

Reviewer 2 Report

Comments and Suggestions for Authors

This paper proposes a semantic laser SLAM combined with a trajectory interpolation method for mobile robot navigation based on the SLAM systems. I think that the paper's quality improves by the additional explanation in the paper.
The following are some suggestions for promoting the understanding of readers to consider:

p.3 L103:
PointNet++ is not the SLAM algorithm framework, it is a framework for semantic segmentation of point cloud.
It is recommended to correct the citation position.

p.11 Fig. 8:
There is no description of Fig. 8 in text. It is recommended to add the description of the Fig. 8.

Fig. 7,8,9:
These figures should add more detailed information, such as split to (a)(b) and a description of each image, for the understanding of readers.

Fig. 10:
Figure 10(b) should add more detailed information such as (c) and Figure 10(c) should add the trajectory path of the vehicle in the figure for the understanding of readers.
Moreover, it is recommended to add the results of the quantitative comparison between the trajectories generated by the proposed algorithm and RTK-GPS data and to show a high degree of spatial alignment clearly.

References:
The reference format is different after [7] in references. They should be correctly unified.

Reviewer 3 Report

Comments and Suggestions for Authors

The authors addressed all my comments.

Author Response

Once again, thank you very much for your comments and suggestions.